# An antifungal polyketide associated with horizontally acquired genes supports symbiont-mediated defense in *Lagria villosa* beetles

Laura V. Flórez[1], Kirstin Scherlach[2], Ian J. Miller[3], Andre Rodrigues [4], Jason C. Kwan [3], Christian Hertweck[2,5] & Martin Kaltenpoth[1]

Microbial symbionts are often a source of chemical novelty and can contribute to host defense against antagonists. However, the ecological relevance of chemical mediators remains unclear for most systems. *Lagria* beetles live in symbiosis with multiple strains of *Burkholderia* bacteria that protect their offspring against pathogens. Here, we describe the antifungal polyketide lagriamide, and provide evidence supporting that it is produced by an uncultured symbiont, *Burkholderia gladioli* Lv-StB, which is dominant in field-collected *Lagria villosa*. Interestingly, lagriamide is structurally similar to bistramides, defensive compounds found in marine tunicates. We identify a gene cluster that is probably involved in lagriamide biosynthesis, provide evidence for horizontal acquisition of these genes, and show that the naturally occurring symbiont strains on the egg are protective in the soil environment. Our findings highlight the potential of microbial symbionts and horizontal gene transfer as influential sources of ecological innovation.

[1] Department for Evolutionary Ecology, Institute of Organismic and Molecular Evolution, Johannes Gutenberg University, Johann-Joachim-Becher-Weg 13, 55128 Mainz, Germany. [2] Department of Biomolecular Chemistry, Leibniz Institute for Natural Products Research and Infection Biology, HKI, Beutenbergstr. 11a, 07745 Jena, Germany. [3] Pharmaceutical Sciences Division, School of Pharmacy, University of Wisconsin, 777 Highland Ave, Madison, WI 53705-2222, USA. [4] Department of Biochemistry and Microbiology, UNESP-São Paulo State University, Av. 24A, n. 1515-Bela Vista, Rio Claro, SP 13506-900, Brazil. [5] Natural Product Chemistry, Friedrich Schiller University, 07743 Jena, Germany. Correspondence and requests for materials should be addressed to L.V.Fór. (email: laflorez@uni-mainz.de) or to K.S. (email: kirstin.scherlach@leibniz-hki.de)

oping with natural enemies is an important driver of evolution. As a result, a broad spectrum of bioactive molecules among chemically defended organisms have been naturally optimized during a dynamic evolutionary interplay with antagonistic partners[1,2]. For animals living in environments like humid soil and leaf litter, substantial exposure to complex and variable pathogen communities can represent a major threat demanding swift and flexible solutions that ensure survival. In such scenarios, defense provided by symbiotic microorganisms might be advantageous due to the potential for higher versatility compared to host-encoded traits. Indeed, defensive symbiosis seems to be a recurrent strategy across diverse animal hosts[3–5]. From an applied perspective, symbiont-produced defensive compounds can be particularly promising due to their prospective compatibility with a eukaryotic host, as well as the opportunity to assess their role within the ecological context of the host. Thus, while microorganisms are long-established as major sources of pharmaceutically relevant molecules, bacterial and fungal symbionts of animals are now considered particularly promising sources of natural products[3,4,6–10].

In contrast to nutritional symbioses, the cost of harboring symbionts in defensive associations might outweigh the benefits in certain contexts due to variation of natural enemy presence in time and space. This translates in symbiotic partnerships that are usually more dynamic in nature, contrasting many of the ancient nutritional symbioses[3]. There is a growing number of reports on bacteria with bioactive potential found in eukaryotic hosts[6,11,12]; however, knowledge on the chemical basis of symbiont-mediated defense in the field is scarce[3,5]. Importantly, this in-depth knowledge is fundamental for understanding the ecological context and evolutionary dynamics of protective symbioses[13–16].

Recently, we have reported on a protective symbiosis between beetles of the tenebrionid subfamily Lagriinae and several coinfecting *Burkholderia gladioli* strains[17,18]. In *Lagria villosa*, we have shown that under controlled laboratory conditions, the symbionts defend the beetle's eggs from fungal infection while present on the egg surface during vertical transmission. In the only culturable strain isolated from the beetles, we identified four secondary metabolites capable of inhibiting fungi and/or bacteria, which are likely responsible for the protective activity of this strain in situ[18]. In the field, however, the cultured strain is rare,

while a so far uncultured strain (*B. gladioli* Lv-StB) is highly abundant and consistently present across individuals[17]. Evidence for environmental acquisition of *B. gladioli* symbionts in addition to the established vertical transmission route in *Lagria*[18], as well as the consistent presence of multiple coinfecting strains[17], suggest a complex and possibly dynamic protective association.

Here, we characterize a novel bioactive polyketide, likely produced by the *B. gladioli* symbiont strain (*B. gladioli* Lv-StB) that is dominant in field-collected *Lagria villosa* beetles. We demonstrate its occurrence in field-collected beetles' eggs, show its ability to inhibit fungal antagonists in vitro and in vivo, describe the chemical structure of the compound, and identify a gene cluster that may be responsible for its biosynthesis, providing evidence for the horizontal acquisition of the cluster.

## Results

**Discovery of a novel bioactive polyketide on *L. villosa* eggs.** Given that the symbiont strain composition in female *L. villosa* beetles was recently shown to differ between laboratory-reared and field-collected individuals[17], we determined the relative abundance of *Burkholderia* strains on beetle eggs laid by field-collected females (Fig. 1), and set out to characterize the bioactive molecules produced by the dominant symbiont strain (*B. gladioli* Lv-StB). Due to the scarcity of the material, a classical preparative bioassay-guided approach by fractionation of a crude extract of *L. villosa* eggs did not seem feasible. Therefore, we used HPLC and mass spectrometry-based micro-fractionation at an analytical scale in combination with an antifungal bioassay to target the active component of the extract. We could identify an antifungal compound with a molecular weight of $m/z$ 749 amu $(M + H)^+$ and a molecular composition of $C_{41}H_{69}N_2O_{10}$ as deduced from high-resolution mass spectrometry measurements (Fig. 2a). Dereplication approaches with natural product databases indicated that the metabolite likely represents a new chemical structure. Knowing the target compound we could then specifically isolate it from approximately 28,000 pooled *L. villosa* eggs at a yield of 600 μg for a full structural elucidation by MS and NMR (Fig. 2b, Materials and Methods, Supplementary Note 1, and Supplementary Figs. 1−7). We thereby identified a new polyketide that we named lagriamide (Fig. 2b). Interestingly, lagriamide

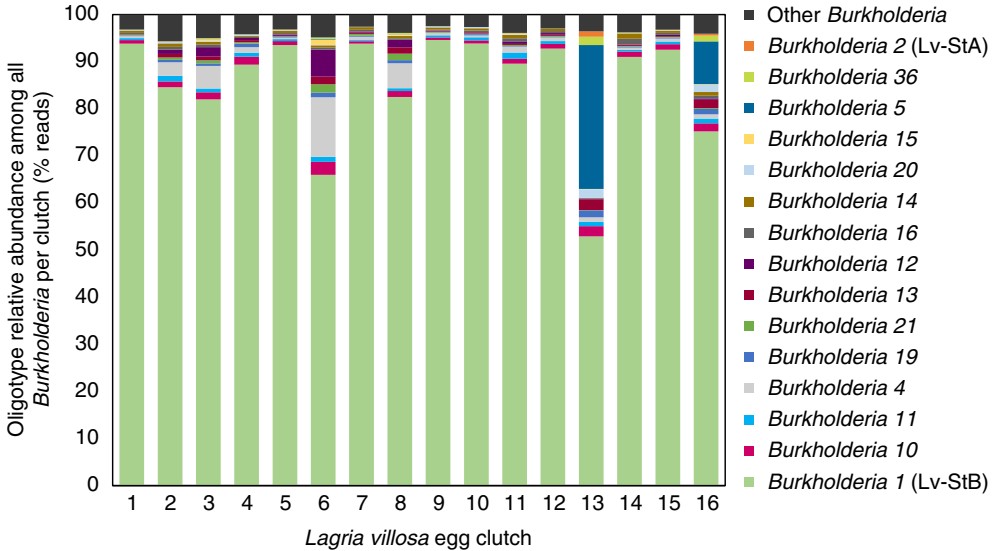

**Fig. 1** *Burkholderia* strain composition on *Lagria villosa* eggs. Relative abundance of *Burkholderia* strains in 16 *L. villosa* egg clutches laid by field-collected females assessed by oligotyping analysis on a 295 bp fragment of the 16S rRNA gene (3467 to 13,677 reads per sample). The assumed correspondence to strains *B. gladioli* Lv-StB and Lv-StA is indicated in parentheses next to the oligotype identifier

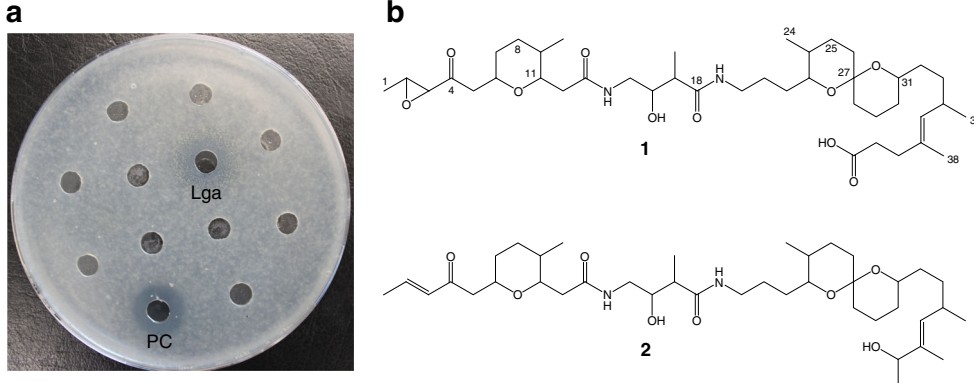

**Fig. 2** The antifungal polyketide lagriamide is present on the eggs of field-collected *Lagria villosa* beetles. **a** Bioassay of fractions from *L. villosa* egg extracts showing antifungal activity against *Aspergillus niger* by the lagriamide-containing fraction (Lga) and nystatin (50 μg mL$^{-1}$) as positive control (PC). **b** Chemical structures of lagriamide (**1**) and bistramide A (**2**)[78]

(**1**) shows similarity to the bistramides (**2**), which are defense compounds of presumed symbiotic origin in didemnid ascidians[19,20].

In addition to the inhibitory activity observed against *Aspergillus niger* in the lagriamide-containing fraction, the general antifungal activity of the pure compound was supported by an in vitro inhibition assay against *Purpureocillium lilacinum* (12 mm inhibition zone around a 9 mm well), a fungus known to infect *L. villosa* eggs in the laboratory[18] and reported as a natural enemy of these beetles[21]. Quantification of lagriamide in egg extracts obtained from independent clutches revealed highly variable amounts, with an average of 40 ng per individual egg (range: 5−97 ng; Fig. 3a and Supplementary Fig. 8). In order to investigate the source of this variation, we quantified the dominant *Burkholderia* symbiont strain in field-collected *L. villosa* (*B. gladioli* Lv-StB) on the same set of egg clutches. Lagriamide was absent in the five clutches lacking this strain, and its abundance significantly correlated with the titers of Lv-StB in all evaluated clutches (Spearman Rank correlation, $p < 0.01$, rho = 0.625, $N = 22$; Supplementary Fig. 8), providing correlative support for strain Lv-StB being responsible for the production of lagriamide. In addition to the egg surface, lagriamide was also detected in extracts from the symbiont-bearing accessory glands of *L. villosa* females from the field (Fig. 3b). This indicates that the production of this compound is not restricted to the egg stage of the host, and that protection on the egg is likely available immediately after oviposition by maternal provisioning of both the symbionts and the protective compound to the egg surface. It is worth noting that a *L. villosa* female has two lagriamide-containing accessory glands, and can lay 3−5 egg clutches during its lifetime, each of a size ranging between 80 and 400 eggs. This is in line with the different orders of magnitude in lagriamide quantities observed in individual eggs and single glands (Fig. 3).

**Putative gene cluster for lagriamide biosynthesis**. Given the previous results suggesting the production of lagriamide by *B. gladioli* Lv-StB and the absence of candidate gene clusters in the whole genome of the cultivated strain (Lv-StA)[18], we set out to investigate whether Lv-StB is indeed responsible for the biosynthesis of this compound. In order to identify the producer, we sequenced the metagenome of eggs laid by field-collected *L. villosa* including their natural microbiota. After de novo assembly of the metagenome reads, we used a binning algorithm where contigs were first separated into kingdom-level bins, then the bacterial contigs were further binned based on coverage, putative taxonomy and 5-mer frequency[22]. This revealed the presence of 20 bacterial bins (Supplementary Table 1), including five that

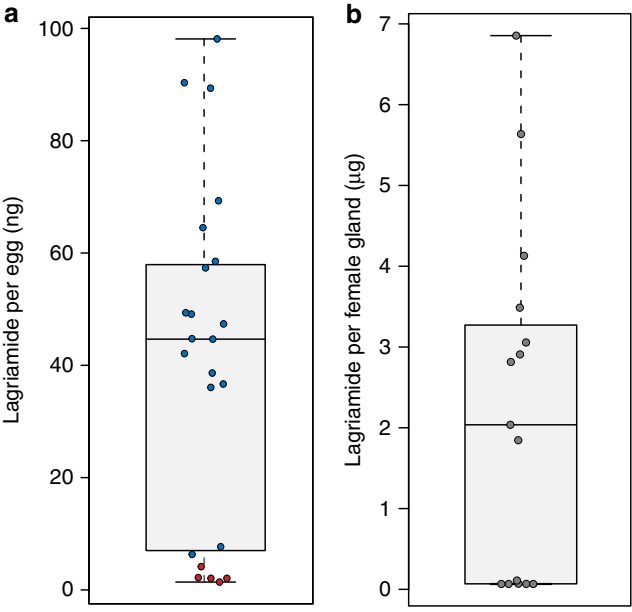

**Fig. 3** Lagriamide is present in variable amounts on eggs, as well as in the female accessory glands of *Lagria villosa* females. **a** Lagriamide amount per egg determined for crude extracts from 22 independent clutches laid by field-collected *L. villosa* females. Circle color indicates the presence (blue) or absence (red) of *B. gladioli* Lv-StB as assessed using qPCR. **b** Lagriamide amount determined for crude extracts from single accessory glands of 15 field-collected *L. villosa* females. The center value of the boxplot represents the median, and the whiskers extend to the most extreme data point which is no more than 1.5 times the interquartile range

were identified as *Burkholderia* strains, and eight further bins in the Burkholderiales. Consistent with previous observations suggesting the predominance of Lv-StB in field-collected *L. villosa* beetles (Fig. 1), the highest coverage bin (DBSCAN_round1_3) from the metagenome, at almost 2000×, contained a contig (NODE_78) with a complete hybrid *trans*-AT- polyketide synthase/non-ribosomal peptide synthetase (*trans*-AT-PKS/NRPS) gene cluster (Fig. 4a).

The putative pathway encoded by this cluster consists of three *trans*-AT PKS genes (*lgaCDG*), two hybrid *trans*-AT PKS/NRPS genes (*lgaAB*) and eight accessory genes. We predict that the pathway is not co-linear, with the last modular protein in the cluster, LgaG, clearly starting with a GCN5-related *N*-

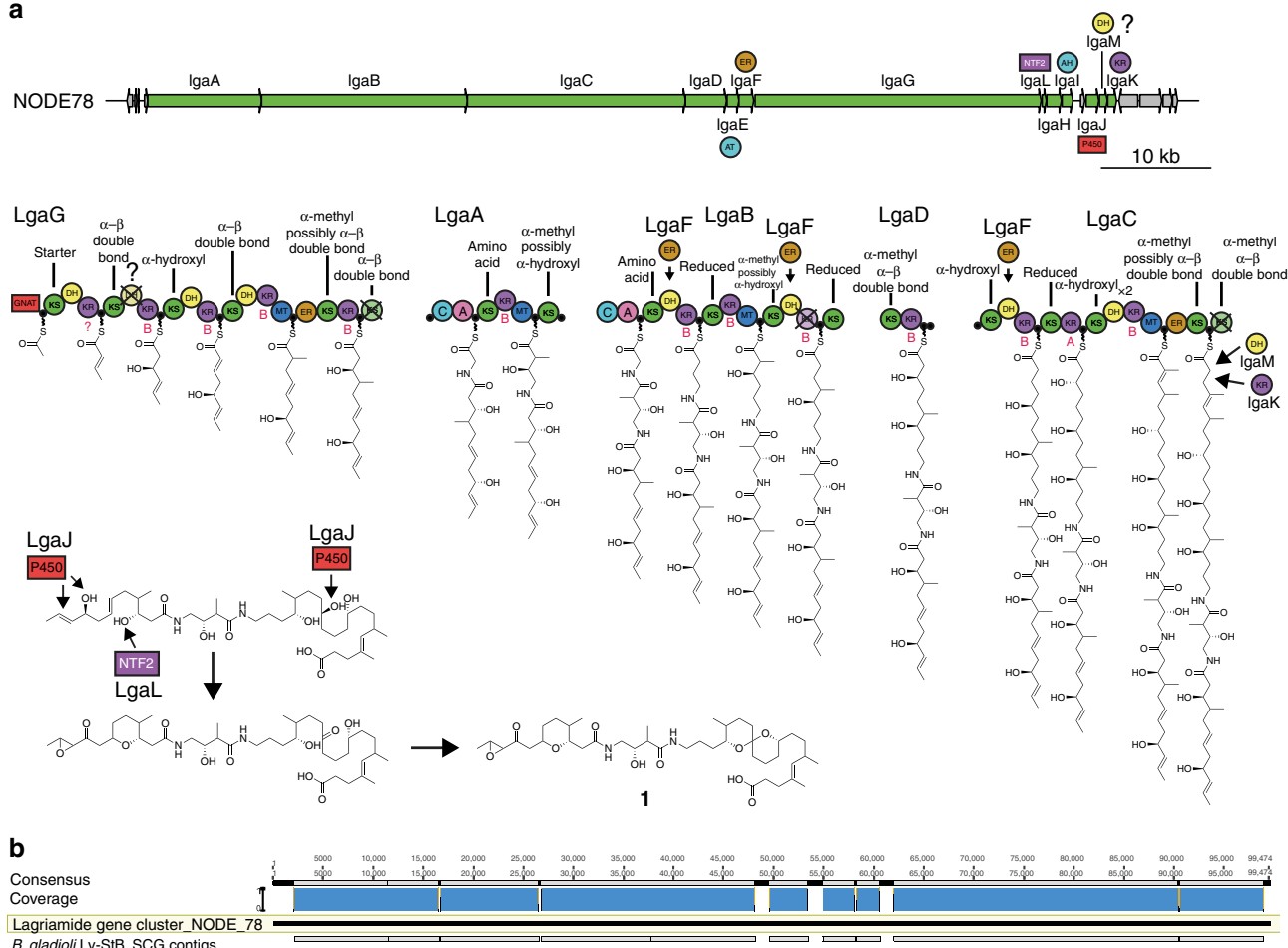

**Fig. 4** A putative gene cluster for lagriamide biosynthesis is present in the genome of the *Burkholderia gladioli* Lv-StB symbionts of *Lagria villosa*. **a** Annotation of the *lgaA-lgaM* hybrid *PKS-NRPS* gene cluster and predicted biosynthesis of lagriamide. The *trans*-AT PKS/NRPS pathway consists of 17 modules. The formation of the epoxide, tetrahydropyran, and spiroacetal moieties is proposed to occur after release from the terminal PKS. Filled black circles depict acyl carrier protein (ACP) domains; a cross indicates domains predicted to be catalytically inactive. Predicted subtype (A or B) is shown adjacent to KR domains in red, and the predicted substrate specificity of KS domains is shown above the respective domain. A: adenylation domain, AT: acyltransferase, AH: acylhydrolase, C: condensation domain, DH: dehydratase, ER: enoyl-reductase, KR: ketoreductase, KS: ketosynthase, MT: *C*-methyltransferase, NTF2: nuclear transport factor 2-like superfamily protein, P450: cytochrome P450. **b** Lagriamide biosynthesis gene cluster aligned to eight of the contigs obtained from single-cell genome sequencing of *B. gladioli* Lv-StB revealing 100% sequence identity

acetyltransferase (GNAT) domain that often loads acetate units onto the first module of *trans*-AT PKS pathways[23,24]. Consistent with the final lagriamide structure, both adenylation domains in NRPS modules are predicted to accept glycine. Similar to other examples of *trans*-AT pathways, we predict the action of enoyl-reductase (ER) LgaF in *trans* for several modules, and this is supported by the predicted substrate specificities of the following ketosynthase (KS) domains based on phylogeny (Supplementary Figs. 9–15)[25]. There are also several KS domains that we predict to be inactive based on the disruption of catalytic residues (Supplementary Fig. 16). We do, however, predict the KS domains at the C-terminals of LgaA and LgaB to be non-extending, even though their catalytic residues appear to be intact. Similar phenomena are predicted in the etnangien[23], lankacidin[24], and patellazoles pathways[26], presumably arising where the next module interacts with the acyl carrier protein (ACP) preceding the active KS. Likewise, alignment of the dehydratase (DH) domains in the cluster (Supplementary Fig. 17) shows the presence of the catalytic histidine and aspartate residues in all cases, which is inconsistent with the presence of an oxygen atom at position 4 in the final molecule. However, the second DH

domain of LgaG appears to have several mutations in the GYxYGPxF and LPFxW motifs that are conserved in the other DH domains of the pathway (Supplementary Fig. 17). These motifs have been found to be widely conserved, except in the case of inactive DH domains[27], and therefore we predict that LgaG_DH2 is inactive. The following KS in module 3 is predicted to accept an α-hydroxyl-containing substrate, supporting this notion. In LgaC, we predict that one module is utilized twice, similar to several other *trans*-AT pathways[23,24] and consistent with the methyl groups at positions 34 and 37 in lagriamide. The C36−C37 double bond could remain due to the ER domain acting on the first iteration only, combined with the substrate specificity of the following KS. The β-position of the last extender unit added by LgaC is predicted to be fully reduced, which may require the in-*trans* action of standalone ketoreductase (KR) LgaK and a DH. BLAST searches of the *lgaM* gene only revealed one hit that was annotated as a hypothetical gene, but it showed weak homology to the "PKS_DH" hidden Markov model (HMM) in the SMART database (accession SM00826). LgaM may therefore be a divergent DH that takes part in the final module of LgaC.

We predict that the epoxide, tetrahydropyran, and spiroacetal groups are not installed by the PKS/NRPS proteins, and the hydroxyl group at position 4 must be oxidized to a carbonyl post-PKS. We speculate that the cytochrome P450 LgaJ is most likely responsible for converting the double bond at C1−C2 to an epoxide, and for oxidizing the hydroxyl at position 4. In some *trans*-AT gene clusters, tetrahydrofuran and tetrahydropyran rings are installed through the action of pyran synthase (PS) domains, that are related to DH domains but possess disrupted active site motifs[28]. However, phylogenetic analysis of the DH domains (Supplementary Fig. 18) in *lga* did not reveal any PS domains, and therefore the tetrahydropyran and spiroacetal groups may be installed post-PKS, or by enzymes acting in *trans*. The gene cluster does, however, encode a protein in the nuclear transport factor 2 (NTF2)-like superfamily (LgaL), which includes $\Delta^5$-3-ketosteroid isomerase, the polyketide cyclase SnoaL[29], and epoxide hydrolases such as MonBI and MonBII[30]. These proteins have been shown to form tetrahydropyran and tetrahydrofuran rings in the monensin[30], indanomycin[31], and salinomycin[32] pathways, among others. We therefore propose that LgaL cyclizes the tetrahydropyran at C6–C11. The spiroacetal group could conceivably be made through a cascade of epoxide-opening reactions similar to the monensin pathway[30], but such a scenario would leave an extra hydroxyl not evident in the final structures. Alternatively, the P450 LgaJ could produce the spiroacetal moiety through oxidation of the central hydroxyl, similar to the process thought to take place in the biosynthesis of the spirangienes[33]. We were able to predict the absolute configurations of several hydroxyl groups based on alignments of KR domains (Supplementary Fig. 19)[34]. However, it is not currently possible to predict the configuration of methylations in *trans*-AT PKS systems, and the configurations of post-PKS processing reactions is uncertain.

In order to complement the metagenomic sequence data and confirm the assignment of the *lga* gene cluster to Lv-StB, six single-cell genomes were obtained from a bacterial suspension of accessory gland contents of two field-collected *L. villosa* females (Supplementary Table 2). The assemblies of all of these genomes contained complete 16S rRNA genes, and two were confirmed as corresponding to Lv-StB. The average nucleotide identity (ANI) of the two single-cell Lv-StB assemblies and DBSCAN_-round_1_3 was found to be 99.985 and 99.97%, indicating significant overlap of these respective genome assemblies. Importantly, all genes from the putative lagriamide biosynthetic gene cluster were found in fragmented form in the single-cell Lv-StB assemblies (Fig. 4b), but in none of the other single-cell genomes, establishing the lagriamide cluster as part of the Lv-StB genome.

The metagenome bin was 2.1 Mbp in size and estimated to be 88.5% complete, based on the presence of single-copy marker genes (Supplementary Table 1), providing a complete genome size estimate of 2.3 Mbp for Lv-StB. This size estimate is supported by the two single-cell genome assemblies, each of which has a size of 1.1 Mbp and is estimated to be 48.3–48.6% complete (Supplementary Table 2). Thus, the Lv-StB genome is significantly smaller than that of the free-living strain Lv-StA (8.5 Mbp) as well as other *B. gladioli* strains (7.8-9.0 Mbp)[35–37], suggesting that Lv-StB has undergone considerable genome reduction.

**Evidence for horizontal acquisition of the gene cluster**. As part of the Autometa algorithm used for metagenomic binning, 5-mer frequencies are reduced to two dimensions with Barnes-Hut stochastic Neighbor Embedding (BH-tSNE)[22]. Inspection of a plot of the two BH-tSNE dimensions against coverage for all contigs in the metagenome (Fig. 5) revealed that the *lga*-containing contig (NODE_78) significantly differed in 5-mer frequency compared to the rest of the Lv-StB genome. Presumably, Autometa had clustered NODE_78 with Lv-StB based on coverage and taxonomic classification—both NODE_78 and the other contigs in Lv-StB are classified as *Burkholderia* and have coverages of almost 2000×, which is between 6- and 240-fold higher than other bins. The occurrence of *lga* genes exclusively in single-cell genomes of Lv-StB suggests that this gene cluster is a single-copy component of the Lv-StB genome rather than a multi-copy component of another genome, as has been observed in another symbiotic system[38]. Local differences in nucleotide composition are a hallmark of horizontal gene transfer events[39], and consistent with this we found transposases at either end of the *lga* gene cluster. Interestingly, analysis of codon adaptation index (CAI)[40] failed to show a significant difference between *lga* genes and other non-hypothetical genes in the Lv-StB genome (Supplementary Fig. 20). Taken together, this suggests that the *lga* gene cluster was horizontally transferred to Lv-StB, and that although codon usage has been normalized to the recipient genome, dicodon bias (as shown with 5-mer frequencies) has not.

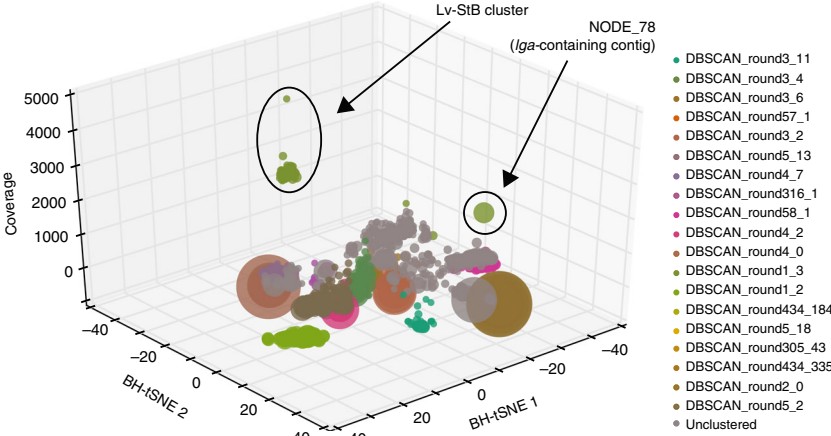

**Fig. 5** The gene cluster putatively involved in lagriamide biosynthesis is present on a genomic island within the *Burkholderia gladioli* Lv-StB genome. 3D scatter plot of de novo shotgun assembly for bacterial contigs greater than 10 kbp from a *Lagria villosa* egg metagenome. Each point represents a contig in the metagenomic assembly and is scaled by length and colored by its corresponding Autometa cluster. The gray points represent contigs that remained unclustered by Autometa

**Symbiont-mediated antifungal activity on eggs exposed to soil**. Previous work on the *Lagria-Burkholderia* association demonstrated symbiont-mediated defense on individual eggs, including controlled infection assays with the culturable strain Lv-StA and the corresponding elucidation of the bioactive compounds toxoflavin, lagriene, caryoynencin, and sinapigladioside[18]. Given our findings on strain diversity and the dominance of strain Lv-StB in field-collected *L. villosa* individuals, we focused on conditions that closely resemble an ecologically relevant setting. To this end, we carried out antifungal assays in field-collected soil, using clusters of eggs from nine different clutches laid by field-collected females within 10 days after collection. Aposymbiotic eggs were generated by egg-surface sterilization with ethanol and bleach[18], and their susceptibility to fungal growth was compared to untreated controls as well as surface-sterilized eggs whose microbiota was subsequently reconstituted by application of an egg wash suspension (reinfected). In a first assay, eggs were exposed to untreated soil, while in a second assay, spores of the egg pathogen *Purpureocillium lilacinum* were inoculated in the soil before coming in contact with the eggs. The level of fungal growth was blindly monitored in all egg clusters after exposure to soil until the eggs hatched, or for the equivalent time (if eggs failed to hatch), and finally compared between the three treatments within each experiment. Both assays revealed significant differences in fungal growth between treatments (Soil assay: Friedman $\chi^2(2) = 10.57$, $p = 0.005$; Soil + *P. lilacinum* assay: Friedman $\chi^2(2) = 7.47$, $p = 0.024$) (Fig. 6). Aposymbiotic egg clusters consistently suffered from higher infection levels in comparison to their untreated symbiotic counterparts, while eggs from the reinfection treatment showed variable levels of fungal growth (Fig. 6).

**Role of lagriamide in antifungal protection on *L. villosa* eggs**. To investigate the role of lagriamide in symbiont-conferred protection, chemical extracts obtained from additional untreated eggs from each clutch were used to evaluate the relationship between the natural abundance of lagriamide and the level of fungal growth on untreated eggs. While fungal growth on eggs exposed to untreated soil did not significantly correlate with lagriamide amounts (Spearmans Rank correlation, $p > 0.05$, rho $= -0.040$), there was a negative correlation in eggs that were exposed to soil inoculated with *P. lilacinum* (Spearmans Rank correlation, $p < 0.05$, rho $= -0.73$) (Supplementary Fig. 21). This assay supports that lagriamide contributes to protection against a specific fungus in the soil environment and suggests that there are likely additional mechanisms, possibly other compounds or strains, involved in the generalized defense of the eggs in the soil.

## Discussion

Beetles of the genus *Lagria* engage in a dynamic defensive symbiosis with multiple coinfecting *B. gladioli* strains. Interestingly, symbiont strain composition seems to be highly sensitive to laboratory conditions, while remaining consistent in the field[17]. This observation stresses the importance of assessing the defensive role of specific strains and the associated chemistry in conditions that closely resemble the natural context, a task that has been challenging in most—if not all—defensive symbioses discovered so far[5].

Here, we identified and characterized the novel polyketide lagriamide, and linked its production to the uncultured *B. gladioli* Lv-StB strain that dominates the egg microbial community in the field. Members of this bacterial genus are recognized as prolific producers of bioactive compounds[41], and the biosynthesis of antifungals by host-associated *Burkholderia* have been reported previously. Rhizoxin, a macrocyclic polyketide, is synthesized by *B. rhizoxinica* symbionts from *Rhizopus microsporus* fungi[42]. Also, a culturable symbiont from *L. villosa* beetles, *B. gladioli* Lv-StA, can produce the isothiocyanate sinapigladioside and the polyyne caryoynencin, which also exhibit antifungal properties[18]. Strikingly, lagriamide is closely related to bistramides (Fig. 2b), a family of compounds isolated from the marine ascidian *Lissoclinum bistratum*[19,20,43]. The structures of bistramides suggest that they are produced by bacterial symbionts as well, although conclusive evidence is currently lacking[44,45]. The structural similarity of lagriamide and bistramides points to a common biosynthetic origin and possible horizontal transfer of the responsible gene cluster, as has been postulated for the symbiotically produced group of compounds comprising onnamide in sponges[46], pederin in staphylinid beetles[47], diaphorin in psyllids[48], and nosperin in lichens[49]. Other defensive compounds in different marine invertebrates, like didemnins and tabjalamins, are also suspected to be encoded in horizontally acquired genes of symbiotic bacteria[4]. Thus, while horizontal gene transfer is known to have an

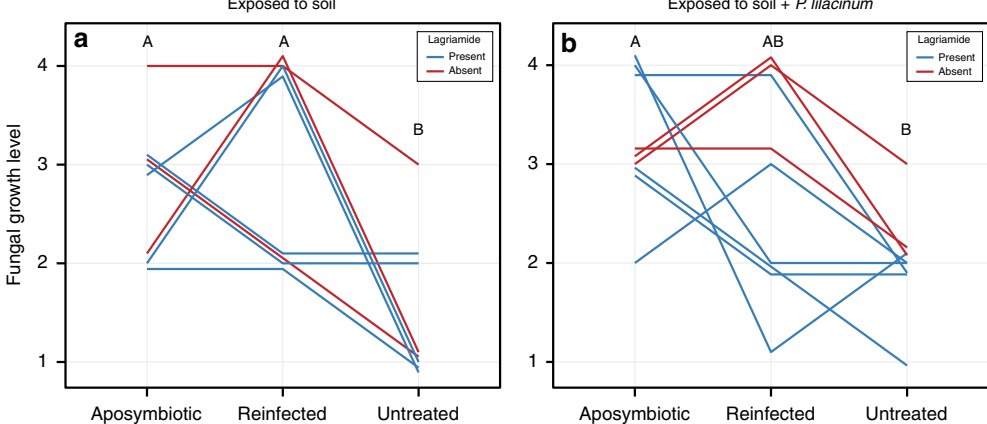

**Fig. 6** Egg clusters lacking the native symbionts are more vulnerable to fungal infection in field-collected soil. **a** Egg clusters exposed to untreated field-collected soil ($N = 8$ egg clutches, Friedman $\chi^2(2) = 10.57$, $p = 0.005$; Nemenyi: A vs. B $p < 0.05$). **b** Egg clusters exposed to field-collected soil inoculated with *P. lilacinum* spores ($N = 9$ egg clutches, Friedman $\chi^2(2) = 7.47$, $p = 0.024$; Nemenyi: A vs. B $p < 0.05$). Fungal growth was estimated qualitatively during blind monitoring of egg clutches using subsets that were either aposymbiotic (symbiont-free), reinfected with a previously recovered egg-wash (reinfected), or untreated (symbiotic), and exposed to soil from their natural environment. The level of growth (1–4) corresponded to increasing amount of fungal biomass visible on the egg surface. Each line corresponds to an independent egg clutch and the color of the line represents absence (red) or presence (blue) of lagriamide in untreated eggs of the corresponding clutch

important impact on the evolution of bacterial genomes[50], we are only starting to understand its putative relevance and prevalence in the specific context of defensive symbiosis. In the *B. gladioli* Lv-StB symbionts of *Lagria*, we additionally found genomic evidence for recent horizontal acquisition of the putative lagriamide gene cluster, supporting that horizontally acquired genes can be important sources of symbiont-conferred defensive traits in animals. Additionally, the co-occurrence of multiple bacterial strains in a shared habitat and a common ecological context, as is the case for the symbionts of *Lagria* beetles, is predicted to facilitate frequent horizontal gene transfer[51].

Interestingly, besides carrying the putative lagriamide gene cluster, the genome of *B. gladioli* Lv-StB strain was noteworthy due to signatures of genome erosion compared to other symbiotic strains of *L. villosa*. While not strongly reduced in comparison to vertically transmitted symbionts of other insects[52], the genome size (around 2.3 Mbp) is estimated to be nearly a fourth of that of *B. gladioli* Lv-StA (8.5 Mbp). The maintenance of a biosynthesis gene cluster encoding for a bioactive compound despite genomic streamlining has been observed in an intracellular bacterial symbiont of the Asian citrus psyllid[48]. In that case there are strong signs of an obligate association, which is rare among defensive symbioses. Similarly, in the Lv-StB symbiont strain of *L. villosa* the reduced genome size might be indicative of long-term vertical transmission and host-symbiont coevolution[53], whereas the larger genome of Lv-StA suggests a more dynamic association with the beetle, likely involving free-living stages.

The coexistence of strains with markedly different genomic features is remarkable given the potentially contrasting evolutionary trajectories associated to their interaction with the insect host. In the context of defensive symbiosis, the presence of coinfecting strains might be beneficial for broad-spectrum protection. As has been observed in the beewolf-*Streptomyces* defensive symbioses, relying on a particular compound or cocktail can be long-term stable and successful against generalist pathogens[15]; however, fluctuating antagonist occurrence or coevolving natural enemies might favor higher versatility. Given the high exposure to a variety of potential microbial antagonists in the soil environment, a strategy combining both reliability and versatility seems plausible, as suggested for the fungus-farming ants and their actinomycete associates[54–56]. In *Lagria* beetles, the co-occurrence of a prevalent symbiont with a reduced genome and multiple—probably horizontally acquired—strains could represent a similarly mixed strategy to cope with a wide variety of antagonists, providing an interesting case to investigate the eco-evolutionary dynamics of defensive symbiosis and symbiont strain coinfections.

The in vivo assays using field-collected soil evaluated the generalized protective role of the egg microbial community against fungi, which was supported by the higher susceptibility to fungal growth in symbiont-free egg clusters albeit a highly variable outcome in reinfected eggs. The inconsistent results in this treatment might be explained by ineffective rates of symbiont re-colonization on the eggs under these conditions. Similar experiments carried out previously in this system using individual eggs exposed to a single antagonistic fungus on sterile filter paper demonstrated that reinfections using egg washes containing the *Burkholderia* symbionts effectively rescued the protective effect[18]. In the present case, however, effective reinfection may have been challenged by several factors inherent to the setup on soil. First, it is likely that the symbiont strains compete with other microorganisms in the soil to re-colonize the egg. Furthermore, the physical characteristics of PBS (presumably lower viscosity than maternal secretions) in combination with the porosity of the soil might cause a significant fraction of the microorganisms in the reinfection suspension to disseminate in the soil rather than to

remain on the egg surface. Although additional protection directly by the maternal secretions on the egg cannot be ruled out, previous findings using second-generation aposymbiotic and symbiotic eggs tease apart this potential effect from symbiont-conferred defense[18]. The assays from our present work lay further evidence of a broad-spectrum antifungal effect by the bacterial associates. Importantly, detecting these effects in egg clusters—as opposed to individual eggs—and in the presence of naturally occurring soil fungal communities provides strong support for the ecological relevance of this defensive symbiosis. We also show that lagriamide exhibits antifungal activity in these semi-natural conditions against *P. lilacinum*. The additional in vitro inhibition of this fungus and *A. niger* further suggests that the antifungal activity is not restricted to a specific fungal antagonist. In combination, our observations indicate that lagriamide, which is likely produced by *B. gladioli* Lv. StB, contributes to protection of *L. villosa* eggs in natural conditions, yet may not be the sole mechanism for anti-pathogen defense.

Protection mechanisms in animals are often complementary, as many use a combination of strategies. In *Lagria* beetles, the presence of antibiotic-producing *B. gladioli* symbionts on the egg surface is an important safeguard for the offspring, which are exposed to a variety of natural enemies in the soil environment. This possibility does not preclude a putative action of other members of the egg microbial community, compounds or physical protection provided directly by the mother, nor that of immune activity on the egg chorion, as has been shown in other insects[57–60]. Embedded in a background of natural complexity, our findings emphasize the value of bacterial symbionts as a source of ecological innovation, especially in the context of rapid and versatile acquisition of genes encoding bioactive secondary metabolites. The symbiont-produced defensive substances of *Lagria* beetles additionally highlight the use of studying defensive symbioses in the search for bioactive compounds with applied value.

## Methods

**Insect collection and rearing.** *L. villosa* individuals were collected in the localities of Itajú, Jaú, and Itirapina within the state of São Paulo, Brazil in January–February 2015 (ICMBio authorization Nr. 45742-1, CNPq process n° 01300.004320/2014-21, IBAMA no. 14BR016151DF). Adults were fed with soybean leaves and kept in ventilated plastic containers at 23−26 °C with a natural light regime. Autoclaved water was supplied in centrifuge tubes with cotton, and moist cotton was provided for egg laying.

**General analytical chemistry procedures.** Analytical HPLC was performed on a Shimadzu LC-10Avp series HPLC system consisting of an autosampler, high-pressure pumps, column oven and PDA. HPLC conditions: C18 column (Euro-spher 100-5, 250 × 4.6 mm) and gradient elution (MeCN/0.1 % (v/v) TFA 0.5/99.5 in 30 min to MeCN/0.1 % (v/v) TFA 100/0, MeCN 100 % for 10 min), flow rate 1 mL min$^{-1}$. Preparative HPLC was performed on a Shimadzu LC-8a series HPLC system with PDA. LC-MS measurements were performed using an Exactive Orbitrap High Performance Benchtop LC-MS with an electrospray ion source and an Accela HPLC system (Thermo Fisher Scientific, Bremen). HPLC conditions: C18 column (Betasil C18 3 μm 150 × 2.1 mm) and gradient elution (MeCN/0.1 % (v/v) HCOOH (H$_2$O) 5/95 for 1 min, going up to 98/2 in 15 min, then 98/2 for another 3 min; flow rate 0.2 mL min$^{-1}$). NMR spectra were recorded on a Bruker AVANCE III 600 MHz instrument equipped with a Bruker cryo platform. The residual solvent signals were used as an internal reference.

**Bioassay-guided fractionation and isolation of lagriamide.** To determine the bioactive compound on *L. villosa* eggs, 156 *L. villosa* egg clutches (containing an estimated 28,000 eggs in total) were extracted with MeOH. Fifty microliters of the crude extract was fractionated via analytical HPLC using an automatic fraction collector for MS analysis and bioactivity determination. Fractions were concentrated using a Speedvac and redissolved in 50 μL MeOH. The bioactivity of each fraction was determined using an agar diffusion assay with *Aspergillus niger* DSM737 as an indicator organism. The active compound was then isolated from the pooled extracts by semipreparative HPLC using a Nucleodur C18HTec column (250 × 10 mm, 5 μm) with a flow rate of 5 mL min$^{-1}$ and a gradient method

(MeCN/H$_2$O 25/75 for 2 min, going up to 100/0 in 20 min). The total yield was 600 µg.

**Structure elucidation of lagriamide.** A molecular mass of $m/z$ 749.4949 amu $(M + H)^+$ and a molecular formula of C$_{41}$H$_{69}$N$_2$O$_{10}$ (calcd. 749.4947) was determined by HRESI-MS. The number of carbon atoms was corroborated by $^{13}$C NMR analysis and the multiplicity was assigned by DEPT135 measurements. Analysis of the H,H-COSY spectra revealed the spin system H14-H17 and the spin system the pyran ring (Supplementary Fig. 1). A chemical shift of δ 94.7 ppm for C-27 pointed to the presence of a spirocyclic ring system which was confirmed by characteristic HMBC couplings (Supplementary Fig. 1). Chemical shifts of δ 53.7 ppm and δ 59.7 ppm for C-2 and C-3, respectively, and a coupling constant of $J_{H,H} = 2$ Hz of the corresponding protons disclosed the epoxide moiety. HMBC couplings of C-4 and H-2/H-3 and H-5/H-6 indicated the connectivity of the partial structures.

**Antimicrobial bioassays.** The bioactivity of lagriamide was assessed by agar diffusion tests against *P. lilacinum* LV1 previously isolated from *L. villosa* eggs[18]. Fifty microliters of the solution (1 mg mL$^{-1}$ in methanol as a stock solution and a respective 1:10 dilution) were filled in agar holes of 9 mm diameter (PDA, seeded with a spore suspension of *P. lilacinum*). The inhibition zone was measured after incubation at 30 °C for 24 h.

**Lagriamide quantification on *L. villosa* eggs and female glands.** Twenty-two *L. villosa* egg clutches laid by field-collected mothers were collected from the rearing boxes, and a subset of eggs per individual clutch (13–40) were extracted in methanol. The remaining eggs were used to set up the in vivo bioassay as described below or kept for rearing. The reproductive system of 15 field-collected adult females was dissected, and a single accessory gland from each female was extracted in methanol. The crude extracts were analyzed by LC-MS, and lagriamide amounts were measured by integration of the peak areas of the extracted mass traces of the ion corresponding to $m/z = 747.4769 – 747.4843$ [M-H]$^-$.

**Quantification of *B. gladioli* Lv-StB on *L. villosa* eggs.** Following methanol extraction of the egg samples used for lagriamide quantification, we recovered the eggs and preserved these in 70 % ethanol until DNA extraction was carried out. For this purpose, the MasterPure$^{TM}$ complete DNA and RNA isolation Kit (Epicentre Technologies) was used as indicated by the manufacturer, including an additional incubation step at 37 °C with 4 µL lysozyme (100 mg mL$^{-1}$) before protein precipitation. The nucleic acids were resuspended in Low TE buffer (1:10 dilution of TE). In order to assess the number of *B. gladioli* Lv-StB 16 S rRNA gene copies on the eggs, the strain-specific primer Burk16S-StB_F (5′-TTGAAGGCTAATATCC TTCAAGA-3′) and *Burkholderia*-specific primer Burk 3.1_R (5′-TRCCATACTC TAGCTTGC-3′) were used for quantitative PCR in a RotorgeneQ cycler (Qiagen) following the protocol described for the Rotor-Gene SYBR Green PCR Kit. PCR conditions were as follows: 95 °C for 10 min, followed by 45 cycles of 95 °C for 10 s and 65 °C for 30 s. A melting curve was subsequently performed with a temperature ramp from 60 to 99 °C within 4.25 min. All amplified products were purified using the InnuPREP PCRpure kit (Analytik Jena) and Sanger sequencing was performed, confirming 100 % identity of the amplicons with the 16S rRNA sequence of *B. gladioli* Lv-StB (GenBank Accession Nr. KU358661).

**In vivo bioassays.** Soil was recovered from five different soybean plantations in which *L. villosa* were collected. Soil samples were homogenized and kept moist by spraying with sterile water until use, which was no later than 10 days after collection. A layer of soil approximately 1 cm thick was added to individual petri dishes (60 × 15 mm) and either nothing (first assay) or 100 µL of a *Purpureocillium lilacinum* spore suspension (20 spores per µl water) (second assay) were added on the soil in the center of each dish. *P. lilacinum* strain LV1[18] was used. The experiment was carried out with nine independent egg clutches laid by *L. villosa* females within the first 10 days after collection from the field. From each clutch, a subset with a known number of eggs was separated for chemical analysis as described above, and the rest of the eggs were divided in three groups of equal size. The first group was left untreated as a control. Eggs from the second and third groups were submerged in sterile PBS and washed gently by pipetting to recover bacterial cells for posterior reinfection of the corresponding set of eggs for each clutch. All washed eggs (second and third group) were then submerged in 70 % ethanol for 5 min, rinsed with sterile water and treated with 12 % NaClO for 30 s. Subsequently, they were washed thoroughly with sterile water, and each of the three groups was divided in half, resulting in a total of six egg clusters (2× aposymbiotic, 2× reinfected and 2× untreated). Each of these clusters was transferred to individual petri dishes either containing moist soil (first assay) or moist soil and *P. lilacinum* spores (second assay). Sterile PBS or the previously recovered wash suspension was added directly to the eggs from the second (aposymbiotic) and third (reinfected) groups, respectively. The volume corresponded to 2.5 µl per egg in both cases. Nothing was added to the control (untreated) group. For one of the clutches in the assay without *P. lilacinum* spores (first assay), no cell suspension was available for reinfection treatment. Therefore, this clutch was excluded from this assay resulting in eight (first assay) and nine (second assay) biological

replicates. The petri dishes were kept at 25 °C within transparent plastic boxes, and the eggs were monitored daily without knowing the group identity of the samples (blind monitoring). The level of fungal growth was assigned to one of the following categories: 0 = no visible growth, 1 = minor growth directly on surface and barely noticeable, 2 = multiple mycelia in contact with surface, 3 = considerable growth on surface, 4 = surface completely covered by mycelia.

**Statistical analysis for in vivo assays.** Fungal growth levels were statistically analyzed using R 3.4.0[61] in RStudio version 1.0.143[62]. To evaluate the effect of treatment on fungal growth, Friedman rank sum tests were used considering each clutch as a separate block to account for the paired design. Post-hoc tests were carried out based on Nemenyi multiple comparisons from the PMCMR package[63]. Parallel plots representing the corresponding results were generated using the ggplot2 package[64]. Spearman rank correlation tests between lagriamide concentration and fungal growth levels for both assays were also done using R 3.4.0.

**DNA isolation and metagenome sequencing.** Eight egg clutches laid by field-collected *L. villosa* were first chemically extracted and preserved using methanol. MeOH was removed completely and the individual samples were homogenized under liquid nitrogen. Subsequently, 300 µL of tissue and cell lysis solution (MasterPure$^{™}$ complete DNA and RNA isolation Kit, Epicentre Technologies) and 4 µL lysozyme (100 mg mL$^{-1}$) were added to each sample. The samples were incubated at 37 °C for 30 min, and the subsequent steps were carried out following the manufacturer's instructions. Isolated nucleic acids were resuspended in Low TE buffer (0.1 × TE) and brought to a concentration of 100 ng µL$^{-1}$. Fifteen microliters aliquots from each of the eight samples were pooled together. In order to remove RNA, the InnuPrep DNA/RNA Mini Kit (Analytik Jena—Biometra) was used. 250 µL of the lysis solution provided in the kit were added to the pooled sample and the subsequent steps were carried out as outlined by the manufacturer. Two elution steps were done using 50 and 30 µL of the provided elution buffer, respectively. The sample was then concentrated in a SpeedVac$^{™}$ for 45 min. The metagenomic DNA was sequenced using Illumina HiSeq (paired end 2 × 250 bp) to a depth of 40 million reads (20.0 Gbp) at the Max Planck Genome Centre (Cologne, Germany).

**Single-cell genome sequencing.** Individual accessory glands were dissected from two field-collected *L. villosa* females, crushed in sterile phosphate buffer saline and stored in 30 % glycerol at −80 °C until use. Samples were thawed, vortexed for 30 s and centrifuged for 30 s at 2000 rpm to remove large particles. The supernatant of each sample was transferred to fresh cryovials, and samples were later pooled and further processed at the Bigelow Laboratory for Ocean Sciences—Single Cell Genomics Center. From the pooled sample, 96 individual bacterial cells were sorted into a microtiter plate and subjected to whole genome amplification using the WGA-X method[65]. The taxonomic affiliation of bacterial cells was elucidated by amplification and sequencing of the 16S rRNA gene using universal primers. Four *B. gladioli* cells (including two Lv-StB samples) and two other bacteria (*Ochrobactrum* and an unidentified Commamonadaceae, later identified as *Acidovorax*) were subjected to shotgun sequencing using Illumina technology[65].

**Metagenome and single-cell genome sequence data analysis.** The metagenomic Illumina reads were assembled de novo with metaSPAdes, version 3.10.0[66]. The resulting contigs were binned using Autometa[22], utilizing the optional steps of initially splitting contigs based on taxonomic kingdom and recruitment of unclassified contigs with supervised machine learning. Scripts within the Autometa package were used to infer the taxonomy of the resulting bins, and to calculate their completeness and purity. Single-cell genomes and metagenomics bins were annotated with Prokka[67], to identify 16S rRNA and protein-coding genes. ANI values were calculated with ANIcalculator[68]. The *lga* pathway was annotated by searching for protein domains with SMART[69] and CDD Search[70]. Active site residues for protein domains were identified through CDD Search, in order to predict domains that were likely to be functionally inactive by virtue of disrupted catalytic residues. KR domains were aligned in ClustalX[71] in order to identify conserved residues that distinguish stereospecific types[34]. The KS tree was constructed from a collection of 666 KS domain sequences from characterized *trans*-AT PKS pathways and the *lga* cluster. Sequences were aligned with Clustal Omega[72], then a tree was made using FastTreeMP[73], with the parameters "-slow –spr 10 –mlacc 3 –bionj –gamma". The resulting tree was visualized with the interactive Tree of Life (iTOL) server[74]. The DH/PS tree was constructed in a similar manner, using examples previously used to classify these domains[28], as well as DH/PS domains from the mandelalides gene cluster[38].

**CAI calculation.** CAI values were calculated according to the formula of Sharp and Li[40], using *lga* genes to calculate relative synonymous codon usage (RCSU) and $w$ values. Genes were categorized as either hypothetical or annotated (any CDS with an annotation by prokka v. 1.12-beta different to "hypothetical"). The CAI values for these gene categories and lagriamide were plotted as box plots in R, using the ggplot2 package[64]. To test for statistically significant differences between groups, one-way analysis of variance (ANOVA) was carried out in R, using the aov function, followed by Tukey's honest significant difference (HSD) test for significance.

**Burkholderia strain composition analysis**. DNA was extracted from 16 *L. villosa* egg clutches laid by field-collected females using the MasterPure™ complete DNA and RNA isolation Kit (Epicentre Technologies), as described above. Bacterial tag-encoded FLX amplicon pyroseqencing (bTEFAP) was performed by an external service provider (MRDNA Lab, Shallowater, TX, USA) using 16S rRNA primers Gray28F (5′-GAGTTTGATCNTGGCTCA-3′) and Gray519R (5′-GTNTTACNGCGGCKGCTG-3′). Sequencing was performed on a Roche 454 FLX based on company protocols. Quality control of the raw sequences was carried out in QIIME[75] and analysis of the *Burkholderia* strain composition was done using oligotyping[76] and minimum entropy decomposition[77], as described previously[17]. All *Burkholderia* reads, as well as reads corresponding to the Burkholderiaceae or Burkholderiales without further taxonomic assignment were included in the strain composition analysis. For noise reduction, each oligotype was required to (i) occur in more than 0.1% of the reads for at least one sample, (ii) represent a minimum of 20 reads in all samples combined, and (iii) have a most abundant unique sequence within the oligotype with a minimum abundance of 20. For graphical representation, oligotypes containing less than 0.5% of the reads per sample were merged into a single category "Other *Burkholderia*".

**Code availability**. Autometa is freely available at https://bitbucket.org/jason_c_kwan/autometa and as a docker image at https://hub.docker.com/r/jasonkwan/autometa under the GNU Affero General Public License 3 (AGPL 3).

**Data availability**. Nucleotide sequence for the biosynthetic gene cluster of lagriamide is available in the NCBI GenBank repository under the accession number MH171092. The full sequence data set including all raw reads from the single-cell genomes and metagenome are available from the corresponding author upon reasonable request. The sequence data used for *Burkholderia* strain composition analysis have been submitted to the NCBI Sequence Read Archive (SRA) database under the BioProject ID PRJNA306502 and accession numbers SAMN04510296 to SAMN04510311.

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

## Acknowledgements

We acknowledge preliminary work in the system carried out by Dr. Claudia Roß, and thank Rebekka Janke for support in insect collection and maintenance. We thank Barbara Urbansky (HKI) for technical support with antimicrobial bioassays. We also thank the Brazilian governmental institutions Instituto Chico Mendes de Conservação da Biodiversidade (ICMBio) and Conselho Nacional de Desenvolvimento Científico e Tecnológico (CNPq) for providing permits. Funding was provided by the Max Planck Society (to L.V.F. and M.K.), the Johannes Gutenberg University (Internal University Research Funding, Stufe I, to L.V.F.), and the Deutsche Forschungsgemeinschaft within the Collaborative Research Center 1127/ChemBioSys (to K.S., C.H.) and the Leibniz Award (to C.H.). This research was performed in part using the computer resources and assistance of the UW–Madison Center for High Throughput Computing (CHTC) in the Department of Computer Sciences. The CHTC is supported by UW–Madison, the Advanced Computing Initiative, the Wisconsin Alumni Research Foundation, the Wisconsin Institutes for Discovery, and the National Science Foundation and is an active member of the Open Science Grid, which is supported by the National Science Foundation and the U.S. Department of Energy's Office of Science.

## Author contributions

L.V.F. and M.K. designed all insect experiments, K.S. and C.H. designed the chemistry work. I.J.M. and J.C.K. designed and performed the metagenomic and genomic analyses. K.S. carried out all analytical chemistry procedures, lagriamide isolation, and in vitro antimicrobial bioassays. L.V.F. carried out sample collection, in vivo bioassays, wet-lab molecular biology work, and the *Burkholderia* strain composition analysis. Experiments and data analyses were carried out with input from M.K., A.R., J.C.K. and C.H. L.V.F. and M.K. wrote the manuscript, and all authors commented on the final draft.

## Additional information

**Competing interests:** The authors declare no competing interests.

