## [Peer Review File · Nature Communications]

REVIEWERS' COMMENTS:

Reviewer #1 (Remarks to the Author):

Florez et al. "An antifungal polyketide associated with horizontally acquired genes supports symbiont-mediated defense in *Lagria villosa* beetles"

The manuscript presents a thorough work of high quality on an interesting defensive insect-microbe symbiosis. The microbial symbiosis in an European *Lagria* species, whose larvae are equipped with peculiar bacteria-filled pouches consisting of invaginated dorsal intersegmental membranes, was reported in as early as 1920s, but since then, the symbiotic system has been almost forgotten by the scientific community for nearly a century. Recently, the authors started working on the *Lagria* symbiotic system, which unveiled a number of novel and insightful aspects of this unique defensive symbiosis (Florez et al. 2017 Nat Com; Florez et al. 2017 Environ Microb). This study is based on these previous studies, but not a mere extension of them but adding several substantial discoveries supported by high quality data. The following findings are of particular importance.

- 1) Identification and characterization of a new symbiont-derived bioactive substance, lagriamide, with defensive role for eggs against fungal pathogens.
- 2) Genomic determination of the uncultivable *Burkholderia* symbiont that is predominantly associated with wild insects but tends to be lost in lab insects.
- 3) Identification of the genes and the pathway for biosynthesis of lagriamide.
- 4) Demonstration of uncultivable and cultivable defensive symbionts coexisting in the same insect populations/species.
- 5) Suggestive cocktail effects and consequences of different defensive substances derived from the coexisting defensive symbiont diversity.

The data are of sufficiently high quality and quantity. The findings constitute a valuable contribution to the research field of general interest. The writing is clear and concise. I have no substantial concerns over the work and the manuscript.

Reviewer #2 (Remarks to the Author):

The manuscript showed the symbiont interaction between *Burkholderia gladioli* bacteria and *Lagria villosa* beetles by emphasizing on the functions of antifungal lagriamide associated with horizontally acquired genes in dominant *B. gladioli* Lv-StB strain from field-collected beetles. The experimental data on a novel antifungal metabolite basis of symbiont defense in field is an interesting result. Due to property of uncultured Lv-StB strain, authors used de novo assembly of metagenome reads for identifying lagriamide biosynthetic gene clusters and also found a considerable small size (~2.3Mbp) of genome compared to this (~8.5Mbp) of another beetle symbiont *B. gladioli* Lv-StA strain.

I have three major questions with related minor questions.

1. Authors suggest that the lag gene cluster was horizontally transferred to Lv-StB through analyses of CAI, local difference in nucleotide composition, and transposases at either end of the lag gene cluster. Based on genome analysis, authors say "horizontally acquired genes of antifungal polyketide supports symbiont-mediated defense in *L. villosa* beetles" in the Title. This kind of phenomenon

between symbiont bacteria and insects is common, but we did not notice it. Or horizontally acquired lag genes in *B. gladioli* Lv-StB for protecting *L. villosa* beetles eggs is extremely unique? Do authors have any evidence on roles of horizontally acquired genes involved in host-microbe interactions including symbiont interactions? Or did authors have any data of horizontally gene transfer of lag gene from other bacteria in soils or within beetle except for genome analyses? What about the distribution of lag genes in other bacteria or microbes? Did author also observe this phenomenon in *Lagria hirta* beetles?

2. What about interactions or relationships between *B. gladioli* Lv-StB and Lv-StA strains? If both strains existed in the field collected beetles, although Lv-StB is dominant in this manuscript, both strains work co-operatively for protecting eggs by producing each antifungal component(s)? What about the population dynamics of Lv-StB and Lv-StA during laboratory rearing time courses?

3. The genome size of Lv-StB strain is about 2.3Mbp. It is a common size for absolute endosymbiont. When genome analyses of Lv-StB, this strain has strong features of absolute endosymbiont such as lack of many metabolic pathways? If Lv-StB strain is absolute endosymbiont, this strain can only survive within beetles? or can survive within plants? I am wondering which host is origin of this Lv-StB strain because this strain is not well adapted in laboratory rearing beetles?

Reviewer #3 (Remarks to the Author):

This interesting article further analyzes an association between beetles and bacteria that Florez and Kaltenpoth have previously studied. In this manuscript they expand these earlier studies on *Burkholderia gladioli* symbionts of *Lagria villosa* beetles. The earlier studies (refs. 17 and 18) had focused on the *B. gladioli* strain Lv-StA, and this study focused on strain Lv-StB. The two strains differ markedly in several respects: They have distinctly different genomes. LvStA's genome was 8.5 Mbp, which is standard for *B. gladioli*; LvStB's genome is a greatly reduced 2.3 Mbp (lines 212-213). They produce different metabolites, and this paper focuses on lagriamide, which is produced by LvStB but not LvStA. And most importantly, LvStA can be cultured while LvStB has not yet been cultured.

This inability to culture LvStB leads to a manuscript that describes a tour de force application of current technology like single cell sequencing, good use of bioinformatic and statistical analyses, and a high level of scholarship and knowledge of the literature. It also leads to a manuscript that is largely circumstantial evidence and plausible hypotheses, not logically tight cases with convincing experimental details.

1. The structure of lagriamide provides a good example of this general effect. Lack of culturability led to a heroic isolation effort from field sources: 28,000 eggs to obtain 600 µg. The ability to get a 2D structure from this amount is impressive, but I'm not sure that the structure determination, which lacks stereochemistry and other details, would meet publication standards for most chemical journals. I don't think that any laboratory could have done better, and I think that the structure is likely correct. With a dozen undefined stereocenters it's unlikely to be an attractive synthetic target, and the real biological activity is unlikely to be investigated in detail.

2. Sequencing was also hampered by the lack of culturability, and the investigators used sophisticated methods to arrive at a 'genome'. But in the end, their binning approach led to an incomplete genome of 2.1 Mbp for an estimated 2.3 Mbp genome.

3. The biosynthetic gene cluster and proposed biosynthesis are concordant with the incomplete

genetic evidence and past studies on PKS clusters. But inability to culture LvStB leaves all of this untested hypotheses.

Conclusion: The argument for publication is that the manuscript points to some good problems for future investigation and represents a a great deal of top notch work, and my recommendation is to publish.

Opinion: The interesting story is likely to why LvStB cannot be (easily) cultured. The authors skirt with this issue (338-9) by hinting that it might be a way to enforce SvStB's association with the beetle. It would be interesting to look hard at the genomes of the two strains to figure out what's missing in LvStB that could be provided by the beetle host. This probably won't work because of the size of the large number of 'missing Mbps', around twice as many as the total Mbps in the known LvStB genome, and the low quality of the genome. Searching for inducers in the beetle might be a better way to go. A somewhat related approach was very useful in unraveling the relationship of a beetle-host interaction in other systems (Current Biology 2010, 20: 69-74).

Reviewer #1 (Remarks to the Author):

Florez et al. "An antifungal polyketide associated with horizontally acquired genes supports symbiont-mediated defense in *Lagria villosa* beetles"

The manuscript presents a thorough work of high quality on an interesting defensive insect-microbe symbiosis. The microbial symbiosis in an European *Lagria* species, whose larvae are equipped with peculiar bacteria-filled pouches consisting of invaginated dorsal intersegmental membranes, was reported in as early as 1920s, but since then, the symbiotic system has been almost forgotten by the scientific community for nearly a century. Recently, the authors started working on the *Lagria* symbiotic system, which unveiled a number of novel and insightful aspects of this unique defensive symbiosis (Florez et al. 2017 Nat Com; Florez et al. 2017 Environ Microb). This study is based on these previous studies, but not a mere extension of them but adding several substantial discoveries supported by high quality data. The following findings are of particular importance.

- 1) Identification and characterization of a new symbiont-derived bioactive substance, lagriamide, with defensive role for eggs against fungal pathogens.
- 2) Genomic determination of the uncultivable *Burkholderia* symbiont that is predominantly associated with wild insects but tends to be lost in lab insects.
- 3) Identification of the genes and the pathway for biosynthesis of lagriamide.
- 4) Demonstration of uncultivable and cultivable defensive symbionts coexisting in the same insect populations/species.
- 5) Suggestive cocktailing effects and consequences of different defensive substances derived from the coexisting defensive symbiont diversity.

The data are of sufficiently high quality and quantity. The findings constitute a valuable contribution to the research field of general interest. The writing is clear and concise. I have no substantial concerns over the work and the manuscript.

>> We appreciate the reviewer's positive feedback.

Reviewer #2 (Remarks to the Author):

The manuscript showed the symbiont interaction between *Burkholderia gladioli* bacteria and *Lagria villosa* beetles by emphasizing on the functions of antifungal lagriamide associated with horizontally acquired genes in dominant *B. gladioli* Lv-StB strain from field-collected beetles. The experimental data on a novel antifungal metabolite basis of symbiont defense in field is an interesting result. Due to property of uncultured Lv-StB strain, authors used de novo assembly of metagenome reads for identifying lagriamide biosynthetic gene clusters and also found a considerable small size (~2.3Mbp) of genome compared to this (~8.5Mbp) of another beetle symbiont *B. gladioli* Lv-StA strain.

I have three major questions with related minor questions.

1. Authors suggest that the lag gene cluster was horizontally transferred to Lv-StB through analyses of CAI, local difference in nucleotide composition, and transposases at either end of the lag gene cluster. Based on genome analysis, authors say “horizontally acquired genes of antifungal polyketide supports symbiont-mediated defense in *L. villosa* beetles” in the Title. This kind of phenomenon between symbiont bacteria and insects is common, but we did not notice it. Or horizontally acquired lag genes in *B. gladioli* Lv-StB for protecting *L. villosa* beetles eggs is extremely unique?

>> In the discussion we have mentioned a few other examples in which there is direct or indirect evidence that bacterial symbionts carry horizontally acquired genes encoding for bioactive compounds (Lines 316-327 in the updated manuscript). This, together with our findings, leads us to argue that it is a potentially common and relevant phenomenon among defensive bacterial symbioses. An explicit sentence on this point has now been included within the same section (Lines 322-324).

Dis authors have any evidence on roles of horizontally acquired genes involved in host-microbe interactions including symbiont interactions?

>> This is an interesting question for further research on the system. However, we consider that it does not fall within the scope of the present manuscript, which focuses on lagriamide production and its significance for the defensive role of the symbionts.

Or did authors have any data of horizontally gene transfer of lag gene from other bacteria in soils or within beetle except for genome analyses? What about the distribution of lag genes in other bacteria or microbes?

>> We mention that highly similar compounds (bistramides) of likely bacterial origin have been found in tunicates (lines 313-317 in updated manuscript). It is therefore possible that orthologous clusters will be identified in other (symbiotic) bacteria in the future. Since there is currently no conclusive information on the presence of *lga* genes in other microorganisms, additional statements on the origin of the *Lga* gene cluster would be largely speculative. Thus, we prefer to refrain from including this in the manuscript.

Did author also observe this phenomenon in *Lagria hirta* beetles?

>> We have not attempted genomic sequencing of unculturable strains from *L. hirta* in order to assess the presence of the lagriamide biosynthesis gene cluster and its horizontal acquisition. However, initial HPLC-MS analyses of extracts from field-collected *L. hirta* beetles provided some first indication on the presence of similar compounds in this system (data not shown). We will address the identification of these compounds in future experiments which are beyond the scope of the present study.

2 What about interactions or relationships between *B. gladioli* Lv-StB and Lv-StA strains? If both strains existed in the field collected beetles, although Lv-StB is dominant in this manuscript, both strains work co-operatively for protecting eggs by producing each antifungal component(s)?

>> It is certainly possible that the multiple co-existing strains contribute to protection, as we discuss in lines 341-353 (updated manuscript). While we do not have direct evidence for

simultaneous protective activity or interaction between these strains in the field, we have shown in a previous paper (ref. 18: Flórez et al. 2017), that Lv-StA produce at least four different bioactive compounds and can also provide antifungal defense on the beetle egg, as we refer to in lines 59-61 of the current manuscript. Further insights on the potential interaction between the strains are desirable, although the inability to culture Lv-StB *in vitro* currently hinders manipulative approaches to address this question.

What about the population dynamics of Lv-StB and Lv-StA during laboratory rearing time courses?

>> While this is a relevant question to further understand the system, the corresponding experiment would require strain quantification covering the full development of field collected individuals (2-3 months). We provide data on Lv-StB titers on individual eggs (Supplementary Fig. 8) and have previously reported on total *Burkholderia* symbiont numbers in eggs, larvae and adult female glands of *L. villosa* (ref. 18: Flórez et al, 2017). We value the suggestion, but considering the provided information and the main purpose of this manuscript, we think that additional time points or information on Lv-StA would not contribute to our conclusions.

3. The genome size of Lv-StB strain is about 2.3Mbp. It is a common size for absolute endosymbiont. When genome analyses of Lv-StB, this strain has strong features of absolute endosymbiont such as lack of many metabolic pathways? If Lv-StB strain is absolute endosymbiont, this strain can only survive within beetles? or can survive within plants?

>> Both a detailed genome analyses and assessing whether Lv-StB survives in a plant host are part of future work. We consider that these two aspects are not essential to support the conclusions on the production, genetic background and functional role of lagriamide in the context of the defensive symbiosis with *L. villosa*.

I am wondering which host is origin of this Lv-StB strain because this strain is not well adapted in laboratory rearing beetles?

>> Previous findings suggest that the *Burkholderia* symbionts of Lagriinae evolved from a plant-associated ancestor (ref. 8: Flórez et al., 2017). However, resolving the evolutionary history of the symbiosis with specific strains is currently challenging. A comprehensive phylogenetic reconstruction using a considerable number of related strains, including multiple symbionts from other Lagriinae beetles, might provide useful information in this regard. Given that these are so far limited, we consider that it goes beyond the scope of the current manuscript.

Reviewer #3 (Remarks to the Author):

This interesting article further analyzes an association between beetles and bacteria that Florez and Kaltenpoth have previously studied. In this manuscript they expand these earlier studies on *Burkholderia gladioli* symbionts of *Lagria villosa* beetles. The earlier studies (refs. 17 and 18) had focused on the *B. gladioli* strain Lv-StA, and this study focused on strain Lv-StB. The two strains differ markedly in several respects: They have distinctly different genomes. LvStA's genome was 8.5 Mbp, which is standard for *B. gladioli*; LvStB's genome is a greatly reduced 2.3 Mbp (lines 212-213). They produce different metabolites, and this paper focuses on lagriamide, which is produced by LvStB but not LvStA. And most importantly, LvStA can be cultured while LvStB has not yet been cultured.

This inability to culture LvStB leads to a manuscript that describes a tour de force application of current technology like single cell sequencing, good use of bioinformatic and statistical analyses, and a high level of scholarship and knowledge of the literature. It also leads to a manuscript that is largely circumstantial evidence and plausible hypotheses, not logically tight cases with convincing experimental details.

1. The structure of lagriamide provides a good example of this general effect. Lack of culturability led to a heroic isolation effort from field sources: 28,000 eggs to obtain 600 µg. The ability to get a 2D structure from this amount is impressive, but I'm not sure that the structure determination, which lacks stereochemistry and other details, would meet publication standards for most chemical journals. I don't think that any laboratory could have done better, and I think that the structure is likely correct. With a dozen undefined stereocenters it's unlikely to be an attractive synthetic target, and the real biological activity is unlikely to be investigated in detail.

>> This is a valid remark. We agree with the reviewer that a detailed assessment of the biological activity of lagriamide would be interesting and likely provide additional insight into the system, however, the inability to culture the strain prevents the isolation of sufficient amounts of the compound at the current stage. Likewise, the synthesis is not a viable option at the moment due to the number of undefined stereocenters as the reviewer already pointed out.

2. Sequencing was also hampered by the lack of culturability, and the investigators used sophisticated methods to arrive at a 'genome'. But in the end, their binning approach led to an incomplete genome of 2.1 Mbp for an estimated 2.3 Mbp genome.

>> These are indeed known challenges of working with unculturable symbionts, which set some constraints on the general conclusions that can be drawn about the Lv-StB symbiont genome. However, we consider that the available genomic data are enough to support the main conclusions of the paper. That is, that lagriamide is produced by Lv-StB and that the corresponding gene cluster has been horizontally acquired.

3. The biosynthetic gene cluster and proposed biosynthesis are concordant with the incomplete genetic evidence and past studies on PKS clusters. But inability to culture LvStB leaves all of this untested hypotheses.

>> We agree with this comment and recognize the current obstacles for providing direct and complete evidence that support the proposed biosynthesis. Based on this comment and following the editor's suggestions, we have toned down the related statements throughout the manuscript.

Conclusion: The argument for publication is that the manuscript points to some good problems for future investigation and represents a great deal of top notch work, and my recommendation is to publish.

Opinion: The interesting story is likely to why LvStB cannot be (easily) cultured. The authors skirt with this issue (338-9) by hinting that it might be a way to enforce SvStB's association with the beetle. It would be interesting to look hard at the genomes of the two strains to figure out what's missing in

LvStB that could be provided by the beetle host. This probably won't work because of the size of the large number of 'missing Mbps', around twice as many as the total Mbps in the known LvStB genome, and the low quality of the genome. Searching for inducers in the beetle might be a better way to go. A somewhat related approach was very useful in unraveling the relationship of a beetle-host interaction in other systems (Current Biology 2010, 20: 69-74).

>> We appreciate the reviewer's suggestions, and will indeed use complementary approaches to address follow-up questions.